# Erastin Inhibits Septic Shock and Inflammatory Gene Expression via Suppression of the NF-κB Pathway

**DOI:** 10.3390/jcm8122210

**Published:** 2019-12-14

**Authors:** Byung Moo Oh, Seon-Jin Lee, Gyoung Lim Park, Yo Sep Hwang, Jeewon Lim, Eun Sun Park, Kyung Ho Lee, Bo Yeon Kim, Yong Tae Kwon, Hee Jun Cho, Hee Gu Lee

**Affiliations:** 1Immunotherapy Research Center, Korea Research Institute of Bioscience and Biotechnology, Yuseong-gu, Daejeon 34141, Korea; bmoh@kribb.re.kr (B.M.O.); hys8520@kribb.re.kr (Y.S.H.); ljw8796@kribb.re.kr (J.L.); espark@kribb.re.kr (E.S.P.); 2Environmental Disease Research Center, Korea Research Institute of Bioscience and Biotechnology, Yuseong-gu, Daejeon 34141, Korea; sjlee@kribb.re.kr (S.-J.L.); limi@kribb.re.kr (G.L.P.); 3Department of Biomolecular Science, KRIBB School of Bioscience, Korea University of Science and Technology (UST), Yuseong-gu, Daejeon 34113, Korea; 4Anticancer Agent Research Center, Korea Research Institute of Bioscience and Biotechnology (KRIBB), Cheongju 28116, Korea; leekh@kribb.re.kr (K.H.L.); bykim@kribb.re.kr (B.Y.K.); 5Protein Metabolism Medical Research Center and Department of Biomedical Sciences, College of Medicine, Seoul National University, Seoul 110-799, Korea; Yok5@snu.ac.kr

**Keywords:** erastin, sepsis, inflammation, NF-κB, ferroptosis

## Abstract

Sepsis is a life-threatening condition that is caused by an abnormal immune response to infection and can lead to tissue damage, organ failure, and death. Erastin is a small molecule capable of initiating ferroptotic cell death in cancer cells. However, the function of erastin in the inflammatory response during sepsis remains unknown. Here, we showed that erastin ameliorates septic shock induced by cecal ligation and puncture or lipopolysaccharides (LPS) in mice, which was associated with a reduced production of inflammatory mediators such as nitric oxide, tumor necrosis factor (TNF)-α, and interleukin (IL)-1β. Pretreatment with erastin in bone marrow-derived macrophages (BMDMs) significantly attenuated the expression of inducible nitric oxide synthase, cyclooxygenase-2, TNF-α, and IL-1β mRNA in response to LPS treatment. Furthermore, we also showed that erastin suppresses phosphorylation of IκB kinase β, phosphorylation and degradation of IκBα, and nuclear translocation of nuclear factor kappa-light-chain-enhancer of activated B cells (NF-κB) in LPS-stimulated BMDMs. Our findings suggest that erastin attenuates the inflammatory response by suppressing the NF-κB signaling pathway, resulting in inhibition of sepsis development. This study provides new insights regarding the potential therapeutic properties of erastin in sepsis.

## 1. Introduction

Sepsis is a life-threatening organ injury that is induced by an abnormal immune response to microbial infection [1]. In general, sepsis is highly associated with an excessive activation of the innate immune system, which includes crucial host protective mechanisms such as inflammation. Inflammation is one of the defense strategies employed to eliminate noxious stimuli, such as microbial infection (viruses and bacteria), tissue injury, and other detrimental conditions [2,3]. Major innate immune cells, namely macrophages and dendritic cells, have pattern recognition receptors (PRRs) that recognize conserved structures among microbial species; these patterns are termed pathogen-associated molecular patterns (PAMPs) and include lipopolysaccharides (LPS), lipoproteins and nucleic acids, and damage-associated molecular patterns (DAMPs) such as uric acid and ATP [4]. After infection, the interactions of various PAMPs and DAMPs with PRRs activate nuclear factor kappa-light-chain-enhancer of activated B cells (NF-κB) signaling, resulting in the induction of inflammation. In most cases, pathogens are eliminated through innate immune responses, but occasional failure to control inflammation can lead to sepsis with organ dysfunction, such as sepsis-associated liver dysfunction (SALD), acute kidney injury (AKI), and sepsis-associated encephalopathy (SAE) [4,5,6].

Sepsis-induced organ damage is caused by an aberrant increase in inflammatory mediators, e.g., nitric oxide (NO), pro-inflammatory cytokines (e.g., interleukin (IL)-1β and tumor necrosis factor (TNF)-α), and glutamate [5,6,7]. During sepsis, macrophages are activated by various stimuli, such as gram-negative bacteria and LPS, resulting in transcriptional upregulation of inflammatory genes, including inducible nitric oxide synthase (iNOS), IL-1β, and TNF-α [7,8]. Although inflammatory responses play a critical role in host defense against infection, excessive activation of innate immune cells may cause severe sepsis [9]. Therefore, inflammation operates as a sophisticated system and anti-inflammatory agents can be effective therapies for inflammatory disorders. In addition to inflammatory molecules, glutamate has a strong connection to sepsis-induced organ dysfunction, including microglia- and macrophage-derived excitotoxicity in the injured central nervous system (CNS) [10,11]; neurotoxic concentrations of glutamate are released from stimulated macrophages (and microglia), dependent on the system Xc^−^ cysteine/glutamate antiporter [12,13,14]. While inhibition of system Xc^−^ might be a potential approach to attenuating sepsis-induced CNS disease, the relationship between system Xc^−^ and inflammation remains unclear.

Erastin was originally described as selectively lethal to human tumor cells with oncogenic RAS mutations [15,16,17]. This small molecule can directly interact with mitochondrial voltage-dependent anion channels 2 and 3 (VDAC2/3) to induce mitochondrial dysfunction, reactive oxygen species (ROS) production, and oxidative cell death [16]. In addition, erastin is a specific inhibitor of system Xc^−^, which exchanges extracellular cysteine and intracellular glutamate across the cell membrane [18]. Since cysteine is necessary for the synthesis of the antioxidant glutathione (GSH), erastin leads to a depletion of intracellular GSH and inactivation of glutathione peroxidase 4 (GPX4), resulting in iron-dependent necrotic cell death, which is termed ferroptosis [19,20]. However, the role of erastin in the inflammatory response during sepsis remains unknown.

In the present study, we aimed to investigate the effect of erastin on acute inflammatory response using in vivo (cecal ligation and puncture (CLP)- and LPS-induced sepsis) and in vitro (LPS-induced in bone marrow-derived macrophages) sepsis models.

## 2. Experimental Section

### 2.1. Animal Studies

C57BL/6 mice were purchased at 6–10 weeks of age from the Korea Research Institute of Bioscience and Biotechnology (Cheongju, Ohchang, Korea). All animal studies were performed in accordance with the guidelines and with the permission of the Korea Research Institute of Bioscience and Biotechnology Institutional Animal Care (KRIBB-AEC-17179, Nov 02, 2017, KRIBB Institutional Animal Care and Use Committee) and Use. LPS (10 mg/kg, in 100 μL PBS) was intraperitoneally (i.p.) injected with or without erastin (20 mg/kg); mice were divided into PBS, LPS, and LPS + erastin groups, *n* = 10. Cecal ligation and puncture (CLP) was performed based on the protocol used in Kim et al. [21]. Briefly, mice were separated into three groups as follows: (1) sham group (*n* = 10); (2) mice administered PBS only after CLP (*n* = 10); (3) mice administered erastin (20 mg/kg) in PBS after CLP (*n* = 10); mice were anesthetized via an intraperitoneal injection of avertin (500 mg/kg). After anesthesia, the abdomens of the mice were shaved and a standard-practice midline incision was made. The cecum was exteriorized and ligated 1 cm from the cecal tip and perforated with a 23 G needle. Next, a small amount of feces was gently squeezed from the perforated site. The cecum was repositioned, and the peritoneum and skin were closed with 6-0 silk sutures. After the surgical procedure, the mice were injected with 1 mL of PBS with or without erastin (20 mg/kg). The mice of the sham group underwent only the laparotomy and cecum exteriorization. DMSO was used as vehicle control.

### 2.2. Plasma Analysis

Blood samples were collected by cardiac puncture of the mice. Serum was obtained by centrifugation of the blood at 13,000× *g* for 30 min at 4 °C. Serum levels of alanine aminotransferase (ALT) and aspartate aminotransferase (AST) were determined using an automatic chemical analyzer (AU480, Beckman Coulter, Brea, CA, USA).

### 2.3. Reagents and Antibodies

LPS, sulfanilamide, *N*-1-napthylethylenediamine dihydrochloride (NED), and DAPI were obtained from Sigma-Aldrich (St. Louis, MO, USA). Erastin was purchased from Calbiochem (San Diego, CA, USA). FIN56 and L-glutamic acid were purchased form Tocris (Minneapolis, MN, USA). The following antibodies were used: iNOS and β-actin (Santa Cruz Biotechnology, Santa Cruz, CA, USA); COX2, IL-1β, TNF-α, IKKβ, p-IKKβ, IκBα, p-IκBα, p65, p-p38, p-p44/42, p-SAPK/JNK, and PARP (Cell Signaling Technology, Danvers, MA, USA); goat anti-rabbit IgG secondary antibody, Alexa Fluor 633 (Thermo scientific, Rockford, IL, USA).

### 2.4. Cells and Cell Culture

For bone marrow-derived macrophage (BMDM) preparation, bone marrow was isolated from the femurs of 6–10 week old C57BL/6 mice (Korea Research Institute of Bioscience and Biotechnology, Cheongju, Korea). Fresh bone marrow cells were plated in DMEM (Gibco, Waltham, MA, USA) plus 10% fetal bovine serum (Gibco), supplemented with 30% L929-conditioned medium as a source of GM-CSF, at 37 °C in a humidified incubator with 5% CO_2_. Fresh medium was added on Day 3–4 and BMDMs were harvested after 7 days of culture. They were then treated with LPS (1 μg/mL) in the presence or absence of various concentrations of erastin.

### 2.5. Measurement of NO Metabolites, PGE_2_, TNF-α, and IL-1β

NO metabolites were quantified using the Griess test. Brifly, Griess reagent was prepared by mixing equal volumes of 0.1% *N*-1-napthylethylenediamine dihydrochloride (NED) and 1% sulfanilamide in phosphoric acid. The culture medium of LPS-stimulated BMDMs with or without erastin was extracted to react with the Griess reagent (1:1). After 5–10 min, absorbance was measured at 540 or 550 nm using a microplate reader (Molecular Devices, Sunnyvale, CA, USA). Serum levels of nitrate and nitrite concentrations were quantified using a Nitrate/Nitrite Colorimetric Assay Kit (Cayman chemical, Ann Arbor, MI, USA). Prostaglandin E_2_ (PGE_2_) concentrations in culture media and mice sera were determined with a Prostaglandin E_2_ enzyme-linked immunosorbent assay (ELISA) Kit (Cayman chemical). TNF-α and IL-1β in culture media and sera were detected using the Duoset ELISA system (R&D systems, Minneapolis, MN, USA). All experiments were performed in accordance with the manufacturers’ instructions.

### 2.6. Western Blot

Cell lysates were obtained using RIPA buffer (100 mM tris-HCl (pH 7.6), 50 mM NaCl, 0.5% NP40, 0.5% sodium deoxycholate, 0.1 mM Na_3_VO_4_, and protease inhibitor cocktail (Sigma)). Proteins from cell lysates were quantified using a Pierce^®^ BCA Protein Assay Kit (Thermo). Next, protein samples (10–50 μg) were subjected to 6%–15% SDS-PAGE and then transferred to PVDF membranes using a Trans-Blot^®^ Turbo^TM^ Transfer pack (Bio-Rad, Hercules, CA, USA). The membranes were incubated with 5% non-fat dry milk in TBS-T for 1 h and probed with the specified antibodies. Antibody-bound proteins were detected by HRP-conjugated secondary antibodies using a Chemiluminescent HRP Substrate (Millipore, Billerica, MA, USA).

### 2.7. Quantitative Real-Time PCR (qPCR)

TRIzol reagent (Thermo) was used to extract total RNA from macrophages. A total of 5 μg RNA was reverse transcribed using a GoScript^TM^ Reverse Transcription System (Promega, Madison, WI) and analyzed using real-time qPCR. Triplicate reactions were performed using the StepOnePlus Real-Time PCR (Thermo). Primer sequences were as follows: β-actin (forward: 5′- TTG CTG ACA GGA TGC AGA AG-3′, reverse: 5′-ACA TCT GCT GGA AGG TGG AC-3′); iNOS (forward: 5′-CAC CTT GGA GTT CAC CCA GT-3′, reverse: 5′-TGG TCA CCT CCA ACA CAA GA-3′); COX2 (forward: 5′- GGC CAT GGA GTG GAC TTA AA-3′, reverse: 5′-ACC TCT CCA CCA ATG ACC TG-3′); IL-1β (forward: 5′-GAG CCC ATC CTC TGT GAC TC-3′, reverse: 5′-AGC TCA TAT GGG TCC GAC AG-3′); TNF-α (forward: 5′-ACG GCA TGG ATC TCA AAG AC-3′, reverse: 5′-AGA TAG CAA ATC GGC TGA CG-3′). RNA quantity was normalized to β-actin content, and gene expression was quantified according to the 2^−ΔCt^ method.

### 2.8. Immunofluorescence Staining

p65 nuclear translocation was detected by immunofluorescence staining using an anti-p65 antibody (Cell signaling Technology) and DAPI (Sigma-Aldrich). BMDMs were seeded on glass coverslips in six well plates and treated with LPS and erastin for 2 h. Cells were washed three times with PBS and fixed with 3.75% formaldehyde in PBS for 10 min. After washing with PBS, the cells were permeabilized with 0.1% saponin in PBS for 10 min and incubated with blocking solution (3% BSA in PBS) for 1 h. After blocking, cells were incubated with the anti-p65 antibody (Cell signaling Technology, Danvers, MA, USA) for 2 h in PBS containing 3% BSA in the dark. The cells were washed three times with PBS and incubated with the anti-rabbit secondary antibody (Alexa Fluor 633; Molecular probe) for 1 h in PBS containing 3% BSA in the dark. After secondary antibody incubation, cells were washed three times with PBS and incubated with DAPI solution (1 μg/mL) for 30 min in the dark. The stained cells were washed three times and mounted using Dako fluorescence mounting medium (Agilent, Santa Clara, CA, USA). Immunofluorescence was analyzed with a ZEISS LSM 510 META confocal microscope. Co-localization of p65 and DAPI was quantified using Pearson’s correlation analysis by ImageJ (National Institutes of Health, Bethesda, MD) program.

### 2.9. Cell Viability Assay

Cell viability was measured using the MTT assay (Thermo fisher scientific, Rockford, IL, USA). In brief, isolated BMDMs plated in 96 well plates (1 × 10^6^ cells/mL) for 16 h were treated with DMSO or erastin in DMSO at the indicated doses with or without 1 μg/mL of LPS stimulation for 24 h. MTT (5 mg/mL in PBS, 10 μL) was then added to each well and incubated at 37 °C for 3 h in the dark. After incubation, the culture medium was replaced with 100 μL DMSO, and the absorbance was quantitated at 570 nm using a multi-well spectrophotometer (Molecular Devices, Sunnyvale, CA, USA).

### 2.10. Reactive Oxygen Species (ROS) Detection Assay

2 × 10^6^ cells BMDMs plated in six well plates were pre-incubated with DMSO or erastin in DMSO for 2 h. After pre-incubation, BMDMs were stimulated with or without 1 μg/mL of LPS for 2 h. Stimulated BMDMs were harvested, then stained with 5 μM of CM-H_2_DCFDA (Thermo scientific, Rockford, IL, USA) for 30 min. After staining, CM-H_2_DCFDA-positive cells were detected using FACSverse (Becton Dickinson, San Jose, CA, USA).

### 2.11. Statistical Analysis

Quantitative data are presented as means ± standard deviation, and significance was analyzed by performing Student’s *t*-test. *p*-values < 0.05 were considered statistically significant.

## 3. Results

### 3.1. Erastin Prevents CLP-Induced Septic Shock Through Suppression of Inflammatory Responses

Sepsis accompanies an excessive inflammatory response that eventually leads to death [22]. To determine whether erastin can affect mortality by acute sepsis in vivo, we used a CLP-induced sepsis mouse model. Mice were randomly separated into three groups as follows: (i) mice that underwent only incision and cecum exteriorization (sham group, *n* = 10); (ii) DMSO as vehicle control was administered to mice after CLP (*n* = 10); (iii) erastin (20 mg/kg) was administered to mice after CLP (*n* = 10). The survival rates of the three groups of mice were monitored for 5 days. Figure 1A shows that the survival rate of the DMSO control group was 20%. Interestingly, the survival rates were dramatically increased in the erastin-injected group (80%) compared to the control group (Figure 1A). In addition, the serum levels of ALT and BUN were decreased in the erastin-treated group, indicating that erastin also suppressed CLP-induced liver and kidney damage (Figure 1B,C). These results suggest that erastin can inhibit CLP-induced septic shock.

Next, to investigate whether erastin could inhibit the CLP-induced inflammatory response, we analyzed the levels of nitric oxide (NO) metabolites, TNF-α, and IL-1β in the serum of mice in the presence or absence of erastin after CLP. Interestingly, reduced serum levels of NO metabolites were found in erastin-injected mice compared to control mice (Figure 2A). Moreover, erastin treatment remarkably decreased serum levels of CLP-induced TNF-α and IL-1β (Figure 2B,C), suggesting that the inflammatory response induced by CLP was attenuated by erastin. Furthermore, iNOS protein levels in the lung, liver, and kidney were significantly decreased after the administration of erastin (Figure 2D–F, Appendix A). Taken together, these findings demonstrate that erastin enhanced the survival of CLP-induced septic mice through suppression of inflammatory responses.

### 3.2. Erastin Inhibits Septic Death and the Inflammatory Response in an LPS-Induced Septic Model

LPS is one of the PAMPs that is recognized by TLR4 and it is known to cause septic shock [23]. To further investigate the survival benefits conveyed by erastin in acute inflammation, we evaluated the effect of erastin on LPS-induced septic mice. To test this, mice were divided into three groups as follows: (i) injected with PBS; (ii) injected with 10 mg/kg of LPS with DMSO as the vehicle control; (iii) injected with 10 mg/kg of LPS with 20 mg/kg of erastin. Similarly to what was observed in the CLP model, the survival rate for erastin-treated mice was significantly higher than that for animals treated with only LPS (Figure 3A). Next, to examine the effect of erastin on the production of inflammatory mediators and cytokines induced by LPS, ELISA was performed on serum samples 24 h after the injection of LPS, with or without erastin treatment. LPS injection promoted the production of NO metabolites. However, erastin treatment attenuated LPS-induced production of NO metabolites (Figure 3B). Accordingly, levels of TNF-α and IL-1β also decreased considerably due to erastin injection (Figure 3C,D). In addition, LPS administration increased the protein expression of iNOS in lung, liver, and kidney tissues compared to those in PBS-only-injected mice. However, this increase by LPS was considerably suppressed by the administration of erastin (Figure 3E–G, Appendix A). These data confirm that erastin prevented the septic shock and inflammatory responses promoted by LPS as well as CLP in vivo.

### 3.3. Erastin Inhibits LPS-Induced Inflammatory Responses in Bone Marrow-Derived Macrophages (BMDMs)

Macrophages have critical roles in inflammatory disorders as they release inflammatory mediators and cytokines in during the inflammatory response [24]. Because erastin suppresses inflammatory responses in septic mice, we examined the anti-inflammatory effect of erastin using BMDMs differentiated from C57BL/6 mice. To determine the optimal concentration of erastin, BMDMs were treated with various concentrations (0, 5, 10, 20, 40, 60, 80, and 100 μM) of erastin for 24 h. Erastin slightly decreased cell viability at concentrations of 60, 80, and 100 μM, while erastin at concentrations below 40 μM did not affect the viability of BMDMs (Appendix A). Therefore, we used erastin at concentrations below 20 μM for subsequent experiments. To investigate whether erastin regulated the expression of iNOS and cyclooxygenase-2 (COX-2), BMDMs were pretreated with indicated concentrations of erastin for 1 h before treatment with 1 μg/mL of LPS. Real-time PCR data showed that LPS induced the mRNA expression of iNOS and COX-2, while erastin pretreatment suppressed these inductions in a concentration-dependent manner (Figure 4A,B). Consistent with these results, erastin inhibited LPS-induced protein expression of iNOS and COX-2 (Figure 4C and Appendix A). Next, we determined the effect of erastin on LPS-induced production of NO metabolites and PGE_2_ in BMDM culture medium. Treatment with LPS resulted in a significant upregulation in the production of NO metabolites and PGE_2_. However, these inductions were markedly suppressed in erastin-treated BMDMs in a concentration-dependent manner (Figure 4D,E), suggesting that erastin attenuated LPS-induced inflammatory mediators by inhibiting expression of iNOS and COX-2 in BMDMs.

To further confirm the anti-inflammatory effect of erastin on BMDMs, we examined mRNA levels of inflammatory cytokines (TNF-α and IL-1β) using real-time PCR. BMDMs stimulated with LPS showed significantly increased expression of TNF-α and IL-1β, while erastin decreased LPS-induced expression of TNF-α (Figure 5A) and IL-1β (Figure 5B) in a dose-dependent manner. Consistent with these results, western blot analysis showed that erastin attenuated LPS-induced protein expression of TNF-α and IL-1β (Figure 5C and Appendix A). We then determined levels of TNF-α and IL-1β in the culture medium by ELISA. Stimulating BMDMs with LPS significantly upregulated TNF-α and IL-1β levels; however, this upregulation was dramatically suppressed through erastin in a dose-dependent (Figure 5D,E) and time-dependent (Figure 5F,G) manner. Taken together, these results indicate that the inflammatory responses of BMDMs induced through LPS-stimulation were suppressed by erastin in vitro.

### 3.4. Erastin Suppresses NF-κB Activation by LPS Stimulation in BMDMs

As NF-κB is a crucial transcription factor that can regulate the expression of various inflammation-induced genes in response to LPS [25], we investigated whether erastin regulated NF-κB activity. To test this, BMDMs were treated with 1 μg/mL of LPS for 15 min in the presence or absence of erastin. Cell lysates were subjected to western blot analysis to assess phosphorylation of IκBα kinase β (IKKβ) in BMDMs. Stimulation with LPS promoted phosphorylation of IKKβ. However, this event was significantly reduced in the presence of erastin (Figure 6A). In addition, erastin attenuated the LPS-induced phosphorylation and degradation of IκBα (Figure 6A and Appendix A). We then investigated whether erastin could affect the nuclear translocation of the NF-κB p65 subunit induced by LPS stimulation. Immunofluorescence staining showed that translocation of the cytosolic NF-κB p65 subunit to the nucleus was enhanced in LPS-stimulated BMDMs compared to that in control cells; however, this translocation was inhibited by erastin (Figure 6B). Western blot analysis verified that stimulation with LPS increased nuclear p65 levels, whereas this event was inhibited by erastin (Figure 6C and Appendix A). Taken together, these results indicate that erastin suppressed LPS-induced NF-κB activation in BMDMs.

## 4. Discussion

Sepsis is a systemic inflammatory response syndrome caused by pathogen infection, leading to organ dysfunction and mortality [26]. Although inflammatory responses by infection are critical for host defense against invading microbes, excessive activation of innate immune cells, such as macrophages, may contribute to sepsis-induced organ dysfunction and lethality [27]. Here, we revealed an important link between inflammation and erastin. We investigated the anti-inflammatory effects of erastin in vivo on CLP- and LPS-induced sepsis. Erastin administration increased the survival rate in mice upon sepsis induction, followed by a reduction in inflammatory mediators and cytokines, including NO, TNF-α, and IL-1β. Furthermore, we extended the analysis in vitro using BMDMs. Pretreatment with erastin attenuated the expression of iNOS, COX-2, TNF-α, and IL-1β in BMDMs stimulated with LPS in a concentration-dependent manner. These findings therefore establish a new function of erastin as a negative regulator of acute inflammation during sepsis.

Erastin is a small molecule capable of initiating ferroptosis, which is an iron-dependent necrotic cell death and is characterized by the accumulation of lipid peroxides [20]. It functionally inhibits the cysteine–glutamate antiporter System Xc^−^ [28]. Although erastin has been shown to promote ferroptotic cell death in various cells, including cancer cells [29], it did not cause cell death in BMDMs (Appendix A). Therefore, ferroptosis induced by erastin may be dependent on cellular context. Consistent with our findings, a previous study reported that erastin promoted proliferation and differentiation in human peripheral blood mononuclear cells (PBMCs) through induction of lipid peroxidation [30], suggesting that lipid peroxidation could be an initiating ferroptotic signal, but not the effector mechanism of cell death. We examined the effect of other ferroptosis inducers, FIN56 (FIN) and glutamate (Glu), on LPS-induced NO production to further validate whether the anti-inflammatory function of erastin was associated with ferroptosis. BMDMs were treated with LPS in the presence of erastin, FIN56, or Glu. Erastin decreased LPS-induced secretion of NO metabolites (Figure 3A), while pretreatment with FIN and Glu did not suppress NO secretion (Appendix A). These findings support the notion that the function of erastin on inflammatory response is not correlated to its role in ferroptosis induction. Erastin can bind to and inhibit mitochondrial VDAC2/3. This binding induces mitochondria dysfunction, ROS production, and oxidative cell death [16]. LPS also induces oxidative stress via ROS generation [23]. Therefore, we investigated whether erastin affected ROS production induced by LPS. Interestingly, treatment with erastin and/or LPS in BMDMs increased intracellular levels of ROS (Appendix A). However, our study and another [30] showed that erastin did not induce cell death. This discrepancy may be attributable to different experimental contexts or cell types. Indeed, this concept is supported by a previous study showing that erastin promotes oxidative cell death in cells expressing activating mutant RAS (BJ-TERT/LT/ST/RAS^V12^), but not in cells expressing wild type RAS (BJ-TERT) [16].

NF-κB is a pleiotropic transcription factor that plays a critical role in regulating the immune response to infection [31]. NF-κB is present in the cytoplasm as an inactivated dimer composed of p65 and p50 subunits. Inflammatory stimuli, such as LPS, promote a series of NF-κB activation pathways: the phosphorylation of the IKK complex, the phosphorylation and degradation of IκB, and release and nuclear translocation of NF-κB [32]. Activated NF-κB promotes the expression of various inflammatory genes, including *iNOS*, *COX-2*, *TNF-α*, and *IL-1β* [33]. In this study, we have provided evidence that erastin suppresses activation of NF-κB and sequentially decreases the expression of inflammatory mediators in macrophages stimulated with LPS (Figure 6). As a recent study showed that mitogen-activated protein kinases (MAPKs) are critical for LPS-induced expression of pro-inflammatory mediators and cytokines [34], we examined the effect of erastin on LPS-induced MAPK activation. BMDMs were treated with LPS in the presence of erastin. Western blot data, using phosphor-specific antibodies against p38, ERK, and JNK, showed that erastin did not affect MAPK activation by stimulation with LPS (Appendix A). Thus, we cannot exclude the possibility that erastin is associated with other signaling pathways; its inhibitory effect on the induction of septic shock and inflammatory response by LPS, at least in part, is attributable to the suppression of the NF-κB pathway. Although the mechanism by which erastin negatively regulates LPS-induced activation of NF-κB is still unclear, one possibility can be found in its inhibition effect of system Xc^−^; a previous study reported that sulfasalazine (SAS), a system Xc^−^ transporter inhibitor, attenuated NF-κB activity by inhibition of IKKβ [35]; however, S-4-carboxy-phenylglycin (S-4-CPG), another inhibitor of system Xc^−^ did not have any effect on NF-κB activity [36]. Therefore, the molecular mechanism of erastin’s inhibitory action on inflammatory response and NF-κB activation requires further investigation in the future.

## 5. Conclusions

Taken together, the present study showed that erastin markedly alleviates CLP- and LPS-induced sepsis in mice. We also showed that erastin attenuates inflammation response, maybe due to suppression of the NF-κB signaling pathway. These novel findings suggest a new therapeutic relevance of erastin to counteracting inflammatory disease. Our in vitro study focused on the effects of erastin on the function of macrophage in sepsis. Since sepsis is a systemic inflammatory response that directly alters both innate and adaptive immune responses, additional studies are needed to determine if erastin has anti-inflammatory effects in other immune cells responsible for sepsis, such as neutrophils and NK cells.

## Figures and Tables

**Figure 1 jcm-08-02210-f001:**
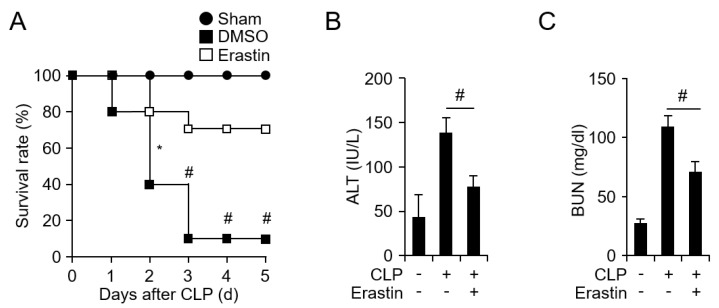
Erastin decreases mortality and organ damage in a CLP-induced sepsis model. (**A**) Mice were subjected to CLP or sham operation, then administrated i.p. injection with or without 20 mg/kg erastin after CLP. All groups, *n* = 10. Mortality of each group was monitored daily for 5 days after surgery operation: sham (circle), DMSO (square), and erastin (empty square). The serum levels of (**B**) alanine aminotransferase (ALT) and (**C**) blood urea nitrogen (BUN) were measured in the sham and CLP model with or without erastin administration. All groups, *n* = 6. Mouse serum samples were collected at 24 h after CLP with or without erastin administration. Graphs represent mean of five independent mice. Statistical analyses were performed using paired two-tailed Student’s *t*-test. * *p* < 0.05, # *p* < 0.01.

**Figure 2 jcm-08-02210-f002:**
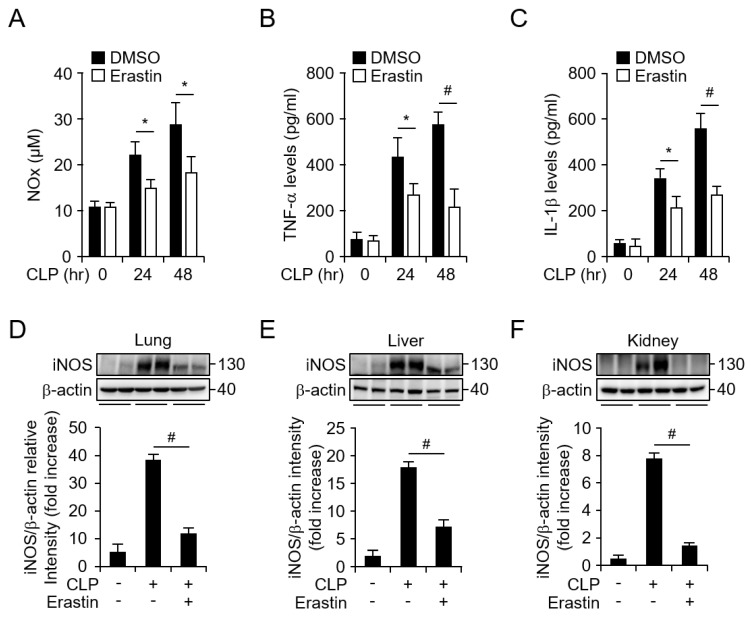
Erastin inhibits NO metabolites (NO_x_), TNF-α, and IL-1β production in a CLP-induced sepsis model. Time-dependent secretion of (**A**) NO metabolites, (**B**) TNF-α, and (**C**) IL-1β with or without erastin treatment after CLP were determined using ELISA assay in mouse serum samples. Mouse sera were collected at 0, 24, 48 h after CLP with or without erastin treatment. All groups, *n* = 5 Graph represents mean of five independent mouse serums level of indicated mediator and cytokines. Protein expression of iNOS in (**D**) lung, (**E**) liver, and (**F**) kidney tissues of sham and CLP-induced sepsis mice with or without erastin treatment were analyzed by western blot. All tissues were randomly collected from two mice of each group at 24 h after CLP with or without erastin administration; β-actin was used as a loading control; graphs represent relative band intensities. Statistical analyses were performed using paired two-tailed Student’s *t*-test. * *p* < 0.05, # *p* < 0.01.

**Figure 3 jcm-08-02210-f003:**
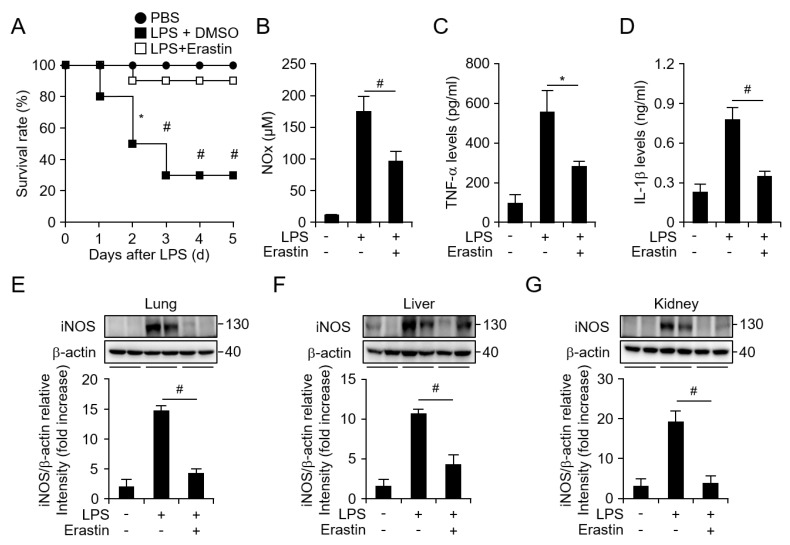
Erastin prevents septic death and downregulates NO metabolites, TNF-α, and IL-1β levels in a mouse model of LPS-induced endotoxemia. (**A**) Graph represents mouse survival rate after PBS and LPS injection (·i.p., 10 mg/kg) with or without erastin treatment (20 mg/kg); PBS (circle), LPS + DMSO (square), LPS + erastin (empty square); all groups, *n* = 10. (**B**) NO metabolites, (**C**) TNF-α, and (**D**) IL-1β secretion after the administration of LPS with or without erastin, as assessed using mouse serum samples. Mouse sera were collected at 24 h after PBS and LPS injection with or without erastin. All groups, *n* = 10. Graphs represent mean of 10 independent mouse serum levels of indicated mediator and cytokines. The expression levels of iNOS in (**E**) lung, (**F**) liver, and (**G**) kidney tissues were analyzed by western blotting after LPS injection with or without erastin treatment. All tissues were randomly collected from two mice of each group at 24 h after LPS injection with or without erastin administration; β-actin was used as a loading control; graphs are shown relative to band intensities. Statistical analyses were performed using paired two-tailed Student’s *t*-test. * *p* < 0.05, # *p* < 0.01.

**Figure 4 jcm-08-02210-f004:**
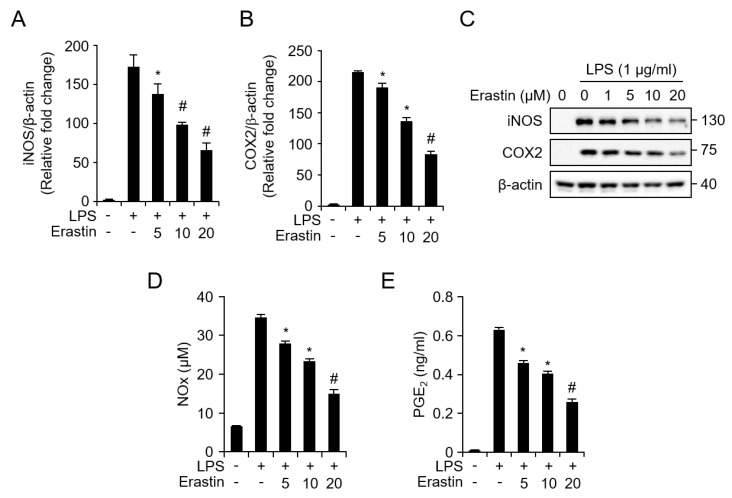
Suppression of LPS-induced NO metabolites and PGE_2_ production by erastin in bone marrow-derived macrophages (BMDMs). (**A**) *iNOS* and (**B**) *COX2* mRNA levels were analyzed by real-time qPCR in BMDMs after stimulation with 1 μg/mL of LPS for 24 h with or without 5, 10, or 20 μM of erastin. (**C**) Protein expression of inflammatory enzymes (iNOS and COX2) in LPS-induced BMDMs with or without the indicated dose of erastin (1, 5, 10, or 20 μM) were examined by western blotting. Cell lysates were harvested at 24 h after LPS treatment with or without indicated dose of erastin; β-actin was used as a loading control. (**D**) NO metabolite secretion was determined using Griess reagent and (**E**) PGE_2_ secretion was analyzed by Prostaglandin E_2_ ELISA Kit in BMDM culture medium. BMDMs were induced with 1 μg/mL of LPS for 24 h with or without erastin at various doses (5, 10, 20 μM). Graphs represent the mean of three independent experiments. Statistical analyses were performed using paired two-tailed Student’s *t*-test. * *p* < 0.05, # *p* < 0.01.

**Figure 5 jcm-08-02210-f005:**
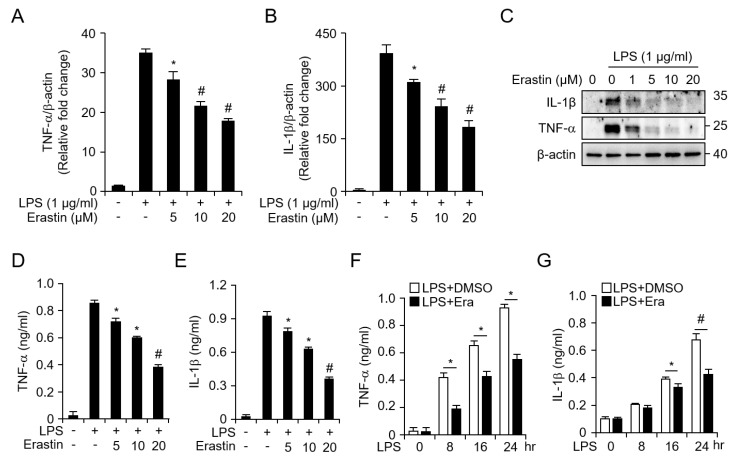
Erastin suppresses the induction of inflammatory cytokines in LPS-induced bone marrow-derived macrophages (BMDMs). Transcription levels of (**A**) TNF-α and (**B**) IL-1β after activation of BMDMs with 1 μg/mL of LPS for 24 h with or without 5, 10, or 20 μM of erastin were measured using real-time qPCR. (**C**) Protein expression of inflammatory cytokines (TNF-α and IL-1β) in LPS-induced BMDMs with or without the indicated dose of erastin (1, 5, 10, or 20 μM) were analyzed by western blot. Cell lysates were harvested at 24 h after LPS treatment with or without indicated dose of erastin; β-actin was used as a loading control. (**D**) TNF-α, and (**E**) IL-1β secretion levels after treating BMDMs with 1 μg/mL of LPS for 24 h with or without erastin at various doses (5, 10, 20 μM) in culture medium were determined by ELISA assay. Time-dependent secretion of (**F**) TNF-α and (**G**) IL-1β from BMDMs following stimulation with 1 μg/mL of LPS for 8, 16, and 24 h with or without 20 μM of erastin was assayed using ELISA. Graphs represent mean of three independent experiments. Statistical analyses were performed using paired two-tailed Student’s *t*-test. * *p* < 0.05, # *p* < 0.01.

**Figure 6 jcm-08-02210-f006:**
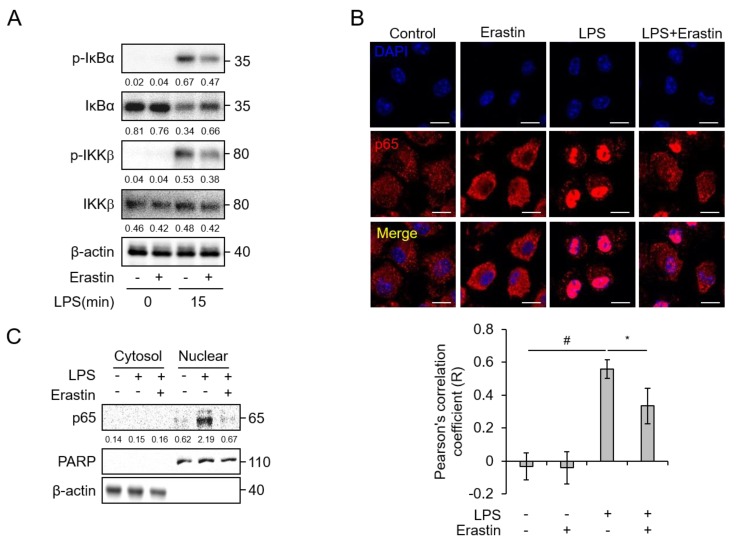
Erastin blocks LPS-induced NF-κB activation in bone marrow-derived macrophages (BMDMs). (**A**) Phosphorylation of IκBα and IKKβ in BMDMs was examined using western blot analysis. BMDMs were pre-incubated with DMSO or erastin in DMSO (20 μM) for 2 h and subsequently inducted with 1 μg/mL of LPS for the indicated times; β-actin was used as a loading control. The numbers below each band indicate quantitative analysis of target protein relative to β-actin. (**B**) Nuclear translocation of NF-κB p65 subunit was detected by immunofluorescence staining in BMDMs after pre-treatment with or without 20 μM of erastin for 2 h, then induction with medium containing 1 μg/mL of LPS for 30 min. Scale bar, 10 μm. Graph represents the co-localization of p65 and DAPI by Pearson’s correlation coefficient. Number of images = 5. (**C**) Western blot analysis of p65 in the cytosol and nuclear fraction of BMDMs with or without pre-incubation with erastin (20 μM), followed by activation with 1 μg/mL of LPS for 30 min; PARP and β-actin were used as nuclear and cytosolic fraction loading controls. The numbers below each band indicate quantitative analysis of target protein relative to β-actin and PARP. Statistical analyses were performed using paired two-tailed Student’s *t*-test. * *p* < 0.05, # *p* < 0.01.

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
