# Peer review of "Erastin Inhibits Septic Shock and Inflammatory Gene Expression via Suppression of the NF-κB Pathway"

_jcm, 2019, doi:10.3390/jcm8122210_

Round 1
Reviewer 1 Report
Oh et al in this study examined the role of Erastin in sepsis-induced inflammatory response. They found that erastin ameliorated septic shock induced by cecal ligation and puncture (CLP) or lipopolysaccharides (LPS) in mice, which was associated with a reduced production of inflammatory mediators. Using BMDMs, they also observed that pre-treatment of BMDMs with erastin dose-dependently attenuated the expression of iNOS, COX2, TNF-α, and IL-1β mRNA in response to LPS treatment. Furthermore, they identifed that erastin suppressed NF-kB signaling pathway.
Overall, this study is interesting and may provide new insights to the treatment of sepsis.
However, there are several minor concerns:
1) Erastin was used in this study prior to CLP or LPS. This means its preventive effects against sepsis. Its therapeutic effects remain unclear. Authors may discuss this as limitation or future study;
2) As authors described in the text, erastin can directly interact with mitochondrial voltage-dependent anion channel 2 and 3 (VDAC2/3) to induce mitochondrial dysfunction, reactive oxygen species (ROS) production, and oxidative cell death. While authors observed that erastin did not induced BMDM death, whether ROS levels are altered in LPS-treated BMDMs by erastin? as LPS could induce oxidative stress.
3) Minor: the subtitle of 3.3 is exactly same as that of 3.2 in the present version, which needs to modify.
Author Response
We are submitting a revised manuscript (jcm-661476) entitled “Erastin inhibits septic shock and inflammatory gene expression via suppression of the NF-κB pathway” for Journal of Clinical Medicine. We greatly appreciate the suggestions and comments of the reviewers. We have addressed their concerns and modified the manuscript accordingly. The modified manuscript was highlighted in red.
Specific comments and responses
Oh et al in this study examined the role of Erastin in sepsis-induced inflammatory response. They found that erastin ameliorated septic shock induced by cecal ligation and puncture (CLP) or lipopolysaccharides (LPS) in mice, which was associated with a reduced production of inflammatory mediators. Using BMDMs, they also observed that pre-treatment of BMDMs with erastin dose-dependently attenuated the expression of iNOS, COX2, TNF-α, and IL-1β mRNA in response to LPS treatment. Furthermore, they identifed that erastin suppressed NF-kB signaling pathway.
Overall, this study is interesting and may provide new insights to the treatment of sepsis.
However, there are several minor concerns:
1. Erastin was used in this study prior to CLP or LPS. This means its preventive effects against sepsis. Its therapeutic effects remain unclear. Authors may discuss this as limitation or future study
This study shows that erastin markedly alleviates CLP- and LPS-induced sepsis in mice. We also show that erastin attenuates inflammation response may be due to suppressing NF-κB signaling pathway in BMDMs. These novel finding provide a new therapeutic relevance of erastin to counteract the inflammatory disease. Our in vitro study focused on the effects of erastin on the function of macrophage in sepsis. Since sepsis is a systemic inflammatory by directly altering both innate and adaptive immune responses, additional studies are needed to determine if erastin has anti-inflammatory effect in other immune cells responsible for sepsis, such as neutrophils and NK cells.
We have now discussed our limitation and future study in Conclusions section.
2. As authors described in the text, erastin can directly interact with mitochondrial voltage-dependent anion channel 2 and 3 (VDAC2/3) to induce mitochondrial dysfunction, reactive oxygen species (ROS) production, and oxidative cell death. While authors observed that erastin did not induced BMDM death, whether ROS levels are altered in LPS-treated BMDMs by erastin? as LPS could induce oxidative stress.
The reviewer makes an excellent point. As reviewer suggested, we investigated whether erastin affect ROS production induced by LPS.
Treatment with erastin and/or LPS in BMDMs can increase in intracellular levels of ROS (Supplementary Fig. 3). However, our and another study (Biochem Biophys Res Commun. 2018, 503, 1689) showed that erastin did not induce cell death. This discrepancy may be attributable to different experimental contexts or cell types. Indeed, this concept is supported by a previous study showing that erastin promotes oxidative cell death in cells expressing activating mutant RAS (BJ-TERT/LT/ST/RASV12), but not in cells expressing wild type RAS (BJ-TERT) (Nature 2007, 447, 864-868).
We have added supplementary Figure 3 and addressed this in discussion section (line 365-373).
3. Minor: the subtitle of 3.3 is exactly same as that of 3.2 in the present version, which needs to modify.
We thank the reviewer for making us aware our error. We made the correction (please see line 254).
We thank the reviewer for insightful and thorough comments and suggestions. We believe that we have addressed their scientific and manuscript concerns. We look forward to a positive assessment of this revised manuscript.

Reviewer 2 Report
Dr. Oh and co-authors present an interesting and solid study showing that the small molecular drug Erastin can inhibit sepsis by reducing the inflammatory response. Concerns are listed below:
Fig 1A, it says n=5 in all groups. Why survival rate shows 70% and 10%? n=? for Figs 1B-C, Figs 2, Figs 3B-G, Figs 4-6. Fig 2A, NOx in the figure, NO in the legend. The time point of Figs 2D-F, Figs 3E-G, Fig 4C, Fig 5C, ? Please show the original plot of WB in Supplementary Data. There is no statistic for Figs 6A and C. It is better to run the correlation analysis for DAPI/p65 images, such as Pearson’s correlation and Manders’ correlation used in Fan, et al. Scientific Reports, 2016. They can be done using FIJI-ImageJ plugins. The in vitro data is focus on macrophages. But it is unclear whether the in vivo data is macrophage dependent. It will be great but not necessary to have a test using Clodronate liposomes. Fig S1, why viability is larger than 100%?Author Response
We are submitting a revised manuscript (jcm-661476) entitled “Erastin inhibits septic shock and inflammatory gene expression via suppression of the NF-κB pathway” for Journal of Clinical Medicine. We greatly appreciate the suggestions and comments of the reviewers. We have addressed their concerns and modified the manuscript accordingly. The modified manuscript was highlighted in red.
Specific comments and responses
Reviewer 2:
Dr. Oh and co-authors present an interesting and solid study showing that the small molecular drug Erastin can inhibit sepsis by reducing the inflammatory response.
Concerns are listed below:
1. Fig 1A, it says n=5 in all groups. Why survival rate shows 70% and 10%?
We performed CLP-induced sepsis experiment (n=5) of two individual experiments. We have corrected it to n=10 (line 187-189).
2. n=? for Figs 1B-C, Figs 2, Figs 3B-G, Figs 4-6.
We appreciate the reviewer for pointing out our omission. We have added the number of samples in Fig 1B-C, Fig 2, Fig 3B-G, Fig 4 and Fig 6
3. Fig 2A, NOx in the figure, NO in the legend.
NO undergoes a series of reactions with several molecules present in biological fluids. The final products of NO are nitrite (NO2-) and nitrate (NO3-). According to reviewer’s comment, we have corrected NO to NO metabolite in main text and legends.
4. The time point of Figs 2D-F, Figs 3E-G, Fig 4C, Fig 5C?
We thank the reviewer for pointing out our omission. You can now find that we updated the Figure legends to address the time points.
5. Please show the original plot of WB in Supplementary Data.
We now show the original blot and mark molecular weight in Supplementary Figure 5~10.
6. There is no statistic for Figs 6A and C.
According to reviewer’s comment, we have now quantified the band intensities in Fig. 6A and C using image J software
7. It is better to run the correlation analysis for DAPI/p65 images, such as Pearson’s correlation and Manders’ correlation used in Fan, et al. Scientific Reports, 2016. They can be done using FIJI-ImageJ plugins.
We thank the reviewer for the supportive comments. We quantified the co-localization of p65 and nuclear (DAPI) using Pearson’s correlation analysis by ImageJ plugins. Based on the Pearson’s correlation analysis, control and erastin group were shown negative correlation (R<0). As expected, LPS+erastin group was shown less positive correlation than LPS group (R value of LPS+erastin group was lower than LPS group). (Cytometry A, 2010, 77 (8), 733-42)
8. The in vitro data is focus on macrophages. But it is unclear whether the in vivo data is macrophage dependent. It will be great but not necessary to have a test using Clodronate liposomes.
Sepsis is a systemic inflammatory by directly altering both innate and adaptive immune responses. Our in vitro study show that erastin attenuates inflammation response may be due to suppressing NK-κB signaling pathway in BMDMs. However, we cannot exclude the possibility that erastin affect other immune cell populations. Therefore, additional studies are needed to determine if erastin has the same effect in other immune cells responsible for sepsis, such as neutrophils and NK cells.
We have now discuss this in Conclusions section.
9. Fig S1, why viability is larger than 100%?
Although erastin has promoted ferroptotic cell death in various cells, including cancer cells, it did not cause cell death in BMDMs (Supplementary Fig. 1). Therefore, ferroptosis induced by erastin may be dependent on cellular context. Consistent with our findings, a previous study reported that erastin promoted proliferation and differentiation in human peripheral blood mononuclear cells (PBMCs) through induction of lipid peroxidation (Biochem Biophys Res Commun. 2018 Sep 10;503(3):1689-1695), suggesting that lipid peroxidation could be an initiating ferroptotic signal, but not the effector mechanism of cell death.
We now explain this more clearly in the discussion section. (Line 353-359)
We thank the reviewer for insightful and thorough comments and suggestions. We believe that we have addressed their scientific and manuscript concerns. We look forward to a positive assessment of this revised manuscript.